# Corneal Epithelial Changes in Diabetic Patients: A Review

**DOI:** 10.3390/ijms25063471

**Published:** 2024-03-19

**Authors:** Lidia Ladea, Mihail Zemba, Maria Ioana Calancea, Mihai Valeriu Călțaru, Christiana Diana Maria Dragosloveanu, Ruxandra Coroleucă, Eduard Lucian Catrina, Iulian Brezean, Valentin Dinu

**Affiliations:** 1Faculty of Medicine, “Carol Davila” University of Medicine and Pharmacy, 050474 Bucharest, Romania; lidia.ladea@gmail.com (L.L.); mhlzmb@yahoo.com (M.Z.); christianacelea@gmail.com (C.D.M.D.); ruxandra.coroleuca@umfcd.ro (R.C.); eduard.catrina@umfcd.ro (E.L.C.); dribrezean@yahoo.com (I.B.); 2Bucharest Emergency Eye Hospital, 030167 Bucharest, Romania; mariaicalancea@gmail.com (M.I.C.); mihaicaltaru7@gmail.com (M.V.C.); 3Ophthalmology Department, “Dr. Carol Davila” Central Military Emergency University Hospital, 010825 Bucharest, Romania; 4Ophthalmology Department, Bucharest Emergency University Hospital, 050098 Bucharest, Romania; 5General Surgery Department, I. Cantacuzino Clinical Hospital, 73206 Bucharest, Romania

**Keywords:** corneal epithelial thickness, AS-OCT, diabetic keratopathy, diabetic corneal neuropathy, IVCM, corneal edema

## Abstract

The relationship between diabetes mellitus and ocular complications has been extensively studied by many authors. Diabetic keratopathy has already been well characterized and defined as a clinical entity. This review focuses on exploring corneal epithelial changes in diabetic patients, aiming to provide a pragmatic overview of the existing knowledge on this topic. The paper systematically examines alterations in corneal epithelial structure and their impact on diabetic patients. Advanced imaging techniques are also discussed for their role in precise characterization and improved diagnostics. Additionally, the paper explores the mechanisms behind corneal epithelial changes in diabetes, looking at factors such as hyperglycemia, oxidative stress, and Advanced Glycation End-Products. The impact of altered corneal epithelial integrity on barrier function and susceptibility to external issues is considered, addressing potential links to heightened proteolytic enzyme activities and delayed wound healing observed in diabetic individuals. The review also covers the practical implications of corneal epithelial changes, including the association with corneal erosions, persistent epithelial defects, and an increased risk of dry eye syndrome in diabetic patients.

## 1. Background

Diabetes mellitus (DM) has been extensively studied for its systemic complications, with a notable impact on ocular health. Diabetic epithelial keratopathy (DEK) represents a distinctive facet of ocular complications in individuals with diabetes, contributing to the growing spectrum of diabetic eye manifestations.

The corneal epithelium serves to protect against environmental factors and to maintain ocular surface integrity and optical clarity. Emerging evidence suggests that diabetes induces notable changes in the corneal epithelium with respect to thickness, cellular morphology, and barrier function. These alterations have been implicated in the pathogenesis of diabetic ocular complications, including dry eye syndrome, recurrent corneal erosions, and delayed wound healing.

The diabetic cornea has long been the subject of research, with many authors reporting on the myriad of clinical and physiopathological phenomena that accompany this disease, such as an increased risk of neurotrophic keratopathy, persistent epithelial defects with or without ulceration, decreased wound healing, decreased basal and wing cell density together with a thickened basement membrane [1,2,3].

As the prevalence of diabetes continues to rise globally, the importance of investigating DEK becomes increasingly apparent. Such studies not only contribute to the fundamental understanding of diabetic ocular complications but also hold the promise of guiding the development of novel therapeutic interventions aimed at preserving corneal health and mitigating vision-related morbidity in individuals with diabetes. This background sets the stage for a comprehensive exploration of corneal epithelial changes in diabetic patients, emphasizing the relevance and urgency of research in this evolving field.

## 2. Clinical Aspects

DEK manifests through a spectrum of clinical signs arising from epithelial dysfunction. A hallmark finding is superficial punctate keratitis (SPK), characterized by fluorescein-positive punctate epithelial erosions. A slit-lamp photograph suggestive of surface changes associated with DEK is shown in Figure 1.

These lesions represent the breakdown of the corneal epithelial surface, often presenting as fine, scattered yellow or white dots on slit-lamp examination. Epithelial fragility and recurrent erosions are another characteristic feature. Diabetic neuropathy can lead to decreased corneal sensitivity, making the eye vulnerable to minor injuries that may go unnoticed. This, coupled with impaired epithelial healing, contributes to recurrent epithelial defects that can be painful and disruptive to vision. These clinical signs highlight the compromised epithelial integrity associated with diabetic keratopathy.

## 3. Central Corneal Epithelial Thickness

Diabetes is recognized for its connection to delayed wound healing, particularly apparent in the corneal epithelium. Patients with diabetes face an increased likelihood of developing conditions such as dry eye, superficial punctate keratitis, recurrent corneal erosion syndrome, and persistent epithelial defects.

Central corneal epithelial thickness (CCET) has been identified in previous research [5] as a possible biomarker of the diabetic cornea. To date, the CCET has been measured using either Anterior Segment Optical Coherence Tomography (AS-OCT), In-Vivo Confocal Microscopy (IVCM) or Very High Frequency Ultrasound (VHF-US), the latter being used exceedingly rarely.

### 3.1. As-Oct-Based Measurements

In recent years, thanks to the advent of Spectral Domain Optical Coherence Tomography (SD-OCT), clinicians have been able to capture images of and measure the thickness of individual corneal layers. Specifically, the corneal epithelium bares mentioning because of its implications in diabetic pathogenesis (Figure 2).

Table 1 shows all studies found at the date of this review that have measured the CCET of diabetic corneas at various disease stages using AS-OCT. D’Andrea et al. [5], found that the central corneal thickness (CCT) was significantly thicker in diabetic patients than in controls, as a result of the corneal epithelial and stromal layers being thicker. Their study population included diabetic patients whose glycemic levels were within the target therapeutic range and no evidence of diabetic retinopathy [5]. In the study, 60 eyes from 30 diabetic patients (16 females and 14 males, with a mean age of 55 ± 4 years) were examined. The average best corrected visual acuity (BCVA) was 0.15 ± 0.2 logMAR, and the mean intraocular pressure (IOP) measured 12.24 ± 1.5 mmHg. The control group consisted of 30 individuals, with an equal distribution of 15 females and 15 males, aged between 49 and 59 years old, resulting in a total of 60 normal eyes, with no underlying conditions. There were no significant differences in terms of age, gender, BCVA, or intraocular pressure (IOP) between the investigated groups. Additionally, there were no significant differences in the results for tear film break-up time (TBUT) and the Schirmer 1 test between the patients and controls. However, CCT was notably higher in diabetic patients (606.26 ± 47.22 m) compared to controls (571.15 ± 42.25 m) (*p* < 0.001). Furthermore, the thicknesses of the corneal epithelial and stromal layers were significantly greater in subjects diagnosed with diabetes when compared to the control group (central sector epithelial thickness: 55.48 ± 3.67 μm vs. 51.80 ± 4.67 μm, *p* = 0.030; central sector stromal thickness: 552.31 ± 45.10 μm vs. 521.93 ± 46.34 μm, *p* < 0.001).

In another investigation conducted by Yusufoğlu et al., it was revealed that individuals with diabetes exhibited a higher mean CCT than those in the control group (*p* = 0.025). However, CCET was lower in the diabetic group (*p* = 0.003) [7]. Their study involved 72 eyes of 72 patients diagnosed with type 2 diabetes, alongside a control group of 72 healthy individuals. The patient group consisted of 43 females and 29 males, with an average age of 57.85 ± 7.83 years, while the control group included 42 females and 30 males, with an average age of 58.83 ± 9.77 years. Among the diabetic patients, the mean disease duration was 12.90 ± 7.51 years, and the mean HbA1c levels were 9.01 ± 1.88. Within the diabetic group, 47.22% had no diabetic retinopathy, 23.61% had non-proliferative diabetic retinopathy (NPDR), and 29.17% had proliferative diabetic retinopathy (PDR). The tear film breakup time and Schirmer test results were also significantly lower in the diabetic group (*p* < 0.001 for both). Notably, these findings remained consistent irrespective of the duration of diabetes and HbA1c levels (*p* > 0.05), and while the presence of retinopathy did not have a significant impact on CCT, it did affect CCET (*p* < 0.001). The authors’ conclusion suggests an inverse relationship between CCET and the severity of diabetic retinopathy.

Examining phacoemulsification surgery candidates, Elmekawey et al. found no statistically significant differences between preoperative measurements of the central, middle and peripheral corneal epithelial thicknesses in diabetic vs. controls [8]. In their diabetic group, there were four patients with type 1 diabetes, and the remaining 21 patients had type 2 diabetes. In terms of HbA1c levels, 30% of the patients had HbA1c levels equal to or greater than 8% (indicating poor control), while 70% had HbA1c levels ranging from 5.5% to 7.5% (indicating good control), with an average of 7.4 ± 1.9%. The average duration of diabetes among these patients was 5.84 ± 4.32 years. One could argue that the diabetes population in this study was shifted towards good glycemic control and a relatively short overall duration of disease, as opposed to the group in the previous study (Yusufoğlu et al. [7]) where the diabetic group had a longer mean duration of disease and higher mean HbA1c levels.

No differences in CCET were found in a pediatric population with DM in a paper by Gunay et al. [9].

### 3.2. IVCM-Based Measurements

As previously stated, IVCM can be used to measure epithelial thickness by subtracting the initial depth from the depth of the scan at the level of the basement membrane. We have found only one paper analyzing epithelial thickness in diabetes using this technology (Table 2). Looking at 44 eyes of 23 patients with diabetes and nine controls using Tandem Scanning Confocal Microscopy (TSCM) (Model 165A, Tandem Scanning Corporation, Reston, VA, USA), Rosenberg et al. found that the epithelial thickness was significantly decreased only in patients with severe neuropathy compared with patients with diabetes without neuropathy (*p* = 0.017) [10].

The main finding of their investigation is that patients with diabetes have no significant differences in epithelial thickness regardless of the systemic complications, unless that complication is severe neuropathy.

The IVCM method of determining epithelial thickness, although accurate, is highly operator dependent, expensive and too cumbersome to be performed routinely, making it unlikely to see wide adoption in ophthalmology clinics moving forward.

## 4. Delayed Wound Healing

Disturbance of trophic factors has been found as being a cause for delayed corneal healing in human and animal diabetic subjects [11,12]. Epithelial defects resulting from diabetic keratopathy, which do not heal, can give rise to significant visual impairments and typically exhibit resistance to standard treatment approaches. This particular aspect of the diabetic cornea can render epithelial thickness measurements highly variable within the same subject at different time points.

Xu et al. showed that high glucose levels induce disruption of the epidermal growth factor receptor (EGFR) pathway in the cornea of diabetic rats and in human corneal epithelial cells, ultimately leading to delayed wound healing [13].

Moreover, simulating diabetic hyperglycemia through high glucose treatment has a rapid and immediate effect on both regular corneal epithelial cells and the epithelium of organ-cultured corneas, leading to reduced cell adhesion and a deceleration in the process of wound healing [14,15,16].

Proteolytic enzymes, such as matrix metalloproteinase (MMP), plasminogen activators and cathepsins, exert significant influence in processes like embryonic development, blood coagulation, wound healing and tissue remodeling. In cases of diabetic corneas marked by delayed and incomplete epithelial wound healing, along with basement membrane fragility and fragmentation, there is a likelihood that proteases play a role in these changes. Indeed, both corneal epithelial cells exposed to high glucose and diabetic corneal epithelium in rats exhibited heightened MMP activity during the wound healing process [17].

The regulation of corneal epithelial wound healing in diabetic individuals is a complex process, guided by a cascade of interactions involving various growth factors and cytokines. This orchestrated cascade leads to the modulation of downstream signaling. The key players in this process, among others, include TGF-β, EGF, HGF, OGF, IGF, NGF, KGF, PDGF, thymosin-β4, IL-6 and IL-10. These factors influence critical aspects such as cell migration, proliferation, differentiation, survival and apoptosis, as extensively discussed in recent articles. The focus in these discussions is on their significance as potential targets for therapeutic interventions in the context of enhancing epithelial wound healing [18,19].

The process of physiological re-epithelialization proves to be more challenging in patients with diabetes, likely attributable to concurrent morphological alterations along the epithelium. These alterations encompass variations in the number of epithelial cell layers, a reduction in endothelial cell count, sectorial thinning, polymorphism, bullae formation, changes in the cellular coefficient of variation and the presence of superficial debris [19].

Another notable use of the in vivo confocal microscopy in the ophthalmogical assessment of patients suffering from diabetes is the ability to evaluate the corneal limbus cells morphology. The corneal pathological changes of the limbus are irregularity of limbal epithelium, which is suffering intraepithelial cystic pathological changes and increased variability of the epithelial cells morphology, resulting in a mosaic pattern, and irregularity of the profound stromal palisades of Vogt, presenting fibrous alterations and disseminated islands of basal limbal epithelial cells. These limbal morphological alterations are believed to be, to some degree, responsible for the delayed corneal wound healing, by affecting the corneal limbal epithelial stem cells and the differentiation and migration of the mature epithelial cells towards the central cornea [20].

## 5. Structural Epithelial Changes

### 5.1. The Epithelial Basement Membrane

A pair of investigations conducted using C57 db/db mice [21,22], along with four separate studies involving rats, revealed that hyperglycemia led to detrimental consequences on the corneal epithelium-basement membrane complex [12,23,24,25]. Moreover, human corneal epithelial cells are shown to exhibit abnormal basement membrane secretion when exposed to hyperglycemia, and there is also evidence that matrix metalloproteinases along with cathepsin F expression and activity are upregulated in the diabetic corneal epithelium [26,27]. This may partly explain the disruption and remodelling of the corneal epithelial basement membrane that has been reported.

### 5.2. Basal Epithelial Cell Density

It appears that in diabetes, the basal corneal epithelial cell density is diminished, both in animal models and in human research studies [28]. Using IVCM, Cai et al. observed decreased basal epithelial cell density (BECD) and reduced thickness of the corneal epithelium in diabetic animal subjects. Ultrastructural analysis reveals that, unlike the normal, columnar-shaped basal epithelial cells, these cells become rounded and contain a large amount of glycogen granules. In addition, one study using scanning electron microscopy revealed abnormal intercellular connections with loss or disturbance of tight junctional complexes between epithelial cells in diabetic rats [29]. These structural transformations are consistent with the malfunction of the epithelial barrier, which is known to have an important contribution in the onset of diabetic keratopathy. Most authors associate these changes with the background of reduced corneal innervation in patients with diabetes, but others suggest that they may not be related to neuropathy unless in severe disease [10,26,30,31,32,33].

### 5.3. The Role of Advanced Glycation End Products

Advanced Glycation End Products (AGEs) are proteins or lipids that suffer the process of glycation after being exposed to sugars. Zou et al. have previously shown that AGEs can accumulate in the cornea [34] and that they trigger the production of reactive oxygen species (ROS) which in turn can lead to cell apoptosis at the epithelial level [25,35,36,37].

AGE accumulation in the corneal epithelium basement-membrane complex renders the latter irregular, thick and multilaminated, thus inducing epithelial damage [25]. On a molecular level, these are associated with biochemical alterations of key structural components, namely a significantly reduced number of hemidesmosomes that also appear to be degenerated and exhibit electron dense deposits [38].

AGEs also have the capacity to induce a non-enzymatic cross-linking within the corneal stroma between the collagen and proteoglycan molecules [34], thereby increasing corneal thickness. Lee et al. have observed this phenomenon in diabetic patients whose duration of disease is greater than 10 years [39].

While many investigations into the biomechanics of the diabetic cornea have been conducted in vivo with human subjects, Bao and colleagues opted to employ rabbits with alloxan-induced diabetes [40]. This choice was influenced by the previously reported biomechanical similarities between rabbit and human corneas. In their research, the authors utilized the tangent modulus (Et) to quantify the elasticity of rabbit corneas, along with the assessment of parameters such as IOP, CCT, blood glucose levels and AGE accumulation in the aqueous humor.

Their study uncovered a significant association between Et, IOP, CCT and blood glucose levels, as well as AGE accumulation. These findings imply that the diabetic cornea experiences heightened stiffness, increased thickness and elevated IOP measurements due to the presence of hyperglycemia and/or the accumulation of AGEs.

## 6. Diabetic Corneal Neuropathy

Corneal neuropathy is a complex and multifaceted condition involving damage and/or dysfunction of the corneal nerves. This condition can result from diabetes as well as herpetic infections or ocular surgeries. Investigating corneal neuropathy is crucial for understanding its pathophysiology, developing effective treatments and improving the quality of life for affected patients. In clinical practice, diabetic corneal neuropathy can be investigated either through imaging (i.e., IVCM) or functionally, through esthesiometry.

### 6.1. IVCM

In vivo confocal microscopy is a non-invasive diagnosis tool, which enables the examiner to directly visualize all of the corneal layers, including the sub-basal nerve plexus (SBNP) (Figure 3). The functioning principle of the confocal microscopy is conjugate orientation of light beams focused on the examined tissue by the condenser lens with the light beams reflected by the tissue gathered by the objective lens. There are three types of in vivo confocal microscopes: tandem-scanning confocal microscope, slit-scanning confocal microscope and laser-scanning confocal microscope, differentiated by their principles regarding light sources and optical systems used within them [41].

In the context of corneal neuropathy, IVCM helps in identifying nerve fiber loss, irregularities in nerve patterns (nerve fiber density, fiber length, branching or beading) and other structural abnormalities. IVCM assessment of these nerve fibers has established itself as an important biomarker regarding the severity of the diabetic neuropathy [43].

IVCM corneal scans on diabetic patients revealed significant alterations such as decreased long nerve fiber bundles in the corneal sub-basal nerve plexus, notable swelling of the sub-basal nerves, as well as decreased sub-basal nerve length and bundle density. All of these pathological changes result in increased corneal sensitivity thresholds in patients with diabetes [20]. Patients with diabetes exhibit corneas with a decreased density of SBNP, a reduction in the number of epithelial nerve fiber bundles per image as observed through IVCM, and increased nerve tortuosity compared to the corneas of individuals without diabetes [44,45].

This damage is primarily responsible for the symptoms encountered by individuals with diabetes suffering from keratopathy, including reduced corneal sensitivity, recurrent corneal erosions, persistent epithelial defects and neurotrophic corneal ulcers. It logically follows that the degree of neuropathy is positively correlated with the delayed wound healing encountered in diabetes, as has been previously shown [14].

### 6.2. Corneal Sensitivity

Lately, corneal sensitivity has been considered as a marker of diabetic neuropathy. In the clinical evaluation of a diabetic patient, this can be measured using several non-invasive tests called corneal esthesiometers, such as Cochet–Bonnet and non-contact corneal esthesiometry (NCCA) devices (e.g., Belmonte) [46].

NCCA devices are considered to have some advantages over Cochet–Bonnet as it is a rapid test with superior stimulus repeatability. An air or gas mixture serves as corneal stimulus for these devices and can assess the sensitivity threshold, which is increased in diabetic patients. The increased sensitivity threshold is associated with modified corneal nerve structure, which may be associated with IVCM findings [46,47].

The Cochet–Bonnet esthesiometer incorporates a nylon monofilament that, in contact with the patient’s cornea, creates a certain amount of pressure which is inversely proportional to the length of the monofilament. As the length is adaptable, the device scale enables the observer to measure the corneal sensitivity threshold [48].

Cotton wisp testing is used to subjectively evaluate corneal sensitivity. The patient is asked to look ahead with both eyes open, and the examiner softly applies the sterile cotton wisp. The examiner repeats the procedure in order to evaluate all four quadrants, in both eyes, comparing the blink responses or the lack thereof. It is less precise compared to the esthesiometers, and it offers no option of quantitative evaluation regarding the corneal sensitivity threshold, but it is extremely useful in physicians’ clinical practice because of its ease of use [49].

Comparing mechanical sensitivity in type I and type II diabetic patients, the thresholds were higher in type II diabetic patients than in type I diabetic patients. A correlation has been observed between the time elapsed since the diabetes diagnosis and the mechanical corneal sensitivity threshold, as both increase proportionally, in type I diabetic patients. No such correlation has been observed in type II diabetic patients [50].

In cases of associated diabetic retinopathy, it is important to keep in mind that laser panretinal photocoagulation procedures can cause a further decrease in corneal sensitivity, as it produces physical injuries to the ciliary nerves [50].

## 7. Central Corneal Thickness

Extensive literature evidence indicates that individuals with type 2 diabetes exhibit higher CCT values compared to healthy controls [51,52,53], with some older studies suggesting that it may be the earliest detectable pathological sign in diabetic individuals [54,55]. This increase in CCT is observed in diabetes patients irrespective of their retinopathy status. Moreover, the severity of peripheral diabetic neuropathy has been associated with a rise in CCT, attributed to an increase in stromal thickness [56].

Numerous studies have demonstrated that hyperglycemia causes damage to the corneal epithelium basement membrane, inducing thickness augmentation [12,21,22,23,24].

Various clinical assessments have explored the relationship between CCT and diabetes, yielding mixed findings. Canan et al. investigated the connection between CCT and diabetic retinopathy status, revealing no significant correlation between CCT and disease duration, stage of retinopathy or prior retinal laser therapy [57]. Conversely, Ozdamar et al. described increased CCT in the PDR group compared to the NPDR group and patients without retinopathy [58]. Comparable outcomes were found in a study by Lee et al., where CCT was notably increased in diabetic patients compared to controls. Moreover, they noticed a correlation between CCT and diabetes duration [39], whereas other authors reported no compelling difference [59,60].

In the context of diabetic epithelial keratopathy, CCT could prove to be a reference point, by using the CCET/CCT ratio, as has been previously studied [61]. Kulikov et al. looked at the fellow eye of pseudophakic bullous keratopathy patients and found that they had a significant reduction in the CCET/CCT ratio without an overall CCT increase, indicative of subclinical stromal thickening secondary to subclinical endothelial dysfunction.

## 8. Corneal Edema

In the absence of endothelial dysfunction, corneal edema secondary to diabetes arises from a disruption of the corneal epithelium’s barrier function, primarily mediated by tight junctions. This disruption leads to increased permeability, allowing the influx of extracellular fluid into the corneal tissue. Elevated glucose levels lead to a decline in the barrier function of corneal and conjunctival epithelial cells. However, this harmful impact is not attributable to a reduction in the levels of tight junction proteins such as claudin-1, zonula occludens-1 (ZO-1), ZO-2, ZO-3 and occludin [62].

The corneal epithelial basement membrane is comprised mainly of tight-junction complexes between epithelial cells, whose main purpose is to act as a barrier, blocking water from infiltrating the stroma [63]. Diabetes can alter the barrier function by damaging the tight-junction complexes, leading to consequent epithelial dysfunction and stromal edema, which is typically rather dehydrated [63,64].

The relationship between diabetes and corneal edema is also influenced by the impact of chronically elevated aqueous glucose levels on the corneal endothelium. This can result in reduced cell density, impairing the endothelium’s ability to effectively regulate corneal fluid balance. As a consequence, individuals with diabetes may be more susceptible to corneal edema. Studies have shown a direct link between diabetes and the corneal endothelial cell density (ECD), especially in the late-stage disease [65,66,67]. There are studies showing that the ECD is decreased in diabetic patients, particularly in type 1 diabetes [65,66].

The reduced ECD along with the epithelial tight-junction complexes damage are responsible for stromal edema, which is an important factor leading to increased corneal thickness [63,64].

## 9. Hysteresis

Diabetes induces transformations of the extracellular matrix of the cornea and thus alters its biomechanical properties [18]. Timely identification of these changes could help screen and monitor ocular complications in patients with diabetes. In clinical practice, there are two measurement devices that can assess corneal biomechanical parameters: the Ocular Response Analyzer (ORA) and the Corvis ST (CST). ORA is a non-contact tonometer that uses a calibrated air-puff and infrared electro-optical system to measure the required force to flatten the cornea as the air pressure rises (P1) and the force at which the cornea becomes flat again as the air pressure falls (P2) [68]. It has great repeatability and reproducibility and it provides two important biomechanical parameters: Corneal Hysteresis (CH) and Corneal Resistance Factor (CRF) [69]. CH reflects the viscoelastic response of the cornea to an applied force defined by a specific air-pressure curve, while CRF gives information on overall resistance of the cornea to deformation. These parameters can change independently, as they are not directly related. CH is calculated as the difference between P1 and P2. CRF is calculated as a linear function of both applanation pressures [68]. CST is also a non-contact tonometer [70] that incorporates a high-speed Scheimpflug camera which records corneal movement during examination. The device creates a two dimensional image of the cross section of the deforming cornea, which is used to measure different parameters such as the amplitude, duration and velocity of the applanation process. These parameters are measured at specific moments of the inward movement (first applanation, A1), the highest concavity state and the backward movement (second applanation, A2) [71].

A systematic review on corneal biomechanics and architecture in patients with diabetes published in 2019 found that most authors report higher CH values in patients with diabetes with a higher corneal viscosity and thus a higher resistance against deformation [18,71]. A cross-sectional observational study on 60 diabetic patients and 48 healthy controls, using ORA as a measurement tool, did not find any differences in CH and CRF between the two groups of patients, nor has it found a correlation between these parameters and the duration of diabetes or serum HbA1c levels. However, it appears that both CH and CRF are positively associated with CCT [68]. There are few studies that use CST for investigation, and their conclusions are somewhat contradictory. Pérez-Rico et al. report significant differences in corneal deformation parameters in diabetic patients while others state that their investigation would rather suggest an unchanged elastic component of corneal behaviour in diabetes [70,72]. As compared to ORA, CST results appear to be more dependent on disease-specific factors such as the coexistence of the diabetic retinopathy and disease duration; although, the associations were quite inconsistent [71].

## 10. Discussion

Clinical signs of DEK include epithelial instability, recurrent erosions, chronic ulcers, corneal edema due to altered epithelial barrier function, superficial punctate keratitis, peculiarly slow and/or incomplete wound healing, reduced epithelial cell density especially in the basal layer, and higher susceptibility to injury [32,73,74,75,76,77]. Most of these changes alter the architecture of the corneal epithelium, including the basement membrane and make interpretation of its imaging challenging.

The normal corneal epithelium (i.e., in a healthy subject) is in a constant state of desquamation and repair, a phenomena also known as the XYZ hypothesis. Within the XYZ hypothesis of corneal epithelial homeostasis, X denotes the proliferation and stratification of limbal basal cells, Y represents the centripetal migration of these basal cells, and Z signifies the desquamation of superficial corneal epithelial cells. In diabetes, the repair arm of the process is impaired and that would theoretically lead to epithelial thinning in all diabetic corneas, but some studies [5] have shown that this is not the case, quite the contrary. Other studies [7,10] have shown that in fact the epithelium is thinner only in cases of advanced diabetes.

The corneal SBNP represents a complex neural network comprising unmyelinated nerve fibers originating from the ophthalmic branch of the trigeminal nerve, densely arrayed just beneath the corneal epithelial basement membrane. The SBNP assumes a pivotal role in maintaining corneal health, encompassing its profound involvement in the context of corneal wound healing and epithelial thickness regulation. Upon injury or insult, neuropeptides and growth factors are released, notably substance P and nerve growth factor (NGF), thereby modulating various wound healing processes. These neuropeptides play a critical role in directing the migration of corneal epithelial cells towards the site of injury, stimulating their proliferation and regulating the inflammatory milieu. Furthermore, the sub-basal nerve plexus contributes significantly to the preservation of the corneal epithelial thickness by fine-tuning the balance between cell proliferation and differentiation. If the epithelium is to be a used as a suitable biomarker for diabetes severity, the SBNP needs to be taken into account. A cornea in a neurotrophic state will have a much more unstable epithelium compared to one with healthy nerves. Therefore, corneal measurements influenced by the epithelium, such as keratometry and CCET may be unreliable, as previous research has shown [78].

Studies performed so far have failed to unequivocally show any sort of correlation between diabetes severity and CCT, which makes the use of this measurement as a biomarker particularly challenging. The lack of statistical significance may stem from confounding factors such as endothelial function or the presence of neuropathy or there may simply be no correlation between the two, regardless of the study design.

Regarding epithelial thickness measurements, in instances where statistical significance was reached and a difference observed, it can be extrapolated that mild/controlled diabetes leads to thickening of the epithelium, whereas advanced/neuropathic diabetes leads to thinning; although, the evidence is not particularly convincing.

Since the intact corneal epithelium plays an important barrier function in preventing stromal edema formation, even a minor reduction in this function will manifest as swelling of the normally relatively dehydrated stroma. The component of the epithelium forming the barrier is largely sub-served by tight junctional complexes between corneal epithelial cells. Loss or disruption of these tight junction structures or loss of basal corneal epithelial cells (as seen on imaging) would explain the loss of the barrier function. The diabetic corneal epithelium can therefore theoretically swell due to the loss of the barrier function and lead to increased thickness measurements. The authors of this paper speculate that it is this subclinical corneal edema found in diabetic corneas with or without endothelial dysfunction that renders epithelial thickness measurements alone quite unreliable as a biomarker for diabetes progression.

A study design that corrects for endothelial dysfunction, corneal neuropathy and assesses epithelial regularity rather than thickness would offer more insight into the matter. Epithelial maps can be acquired using the newer AS-OCT machines and would be more sensitive to changes in epithelial homeostasis, while also being somewhat independent of any thickness changes secondary to edema, which would appear as more generalized thickening. Further studies are necessary in order to assess if indeed epithelial regularity is significantly altered in various stages of diabetes and to what extent. If so, a useful biomarker in this scenario would be a variable that would quantify epithelial thickness homogenity, rather than simply using CCET.

## Figures and Tables

**Figure 1 ijms-25-03471-f001:**
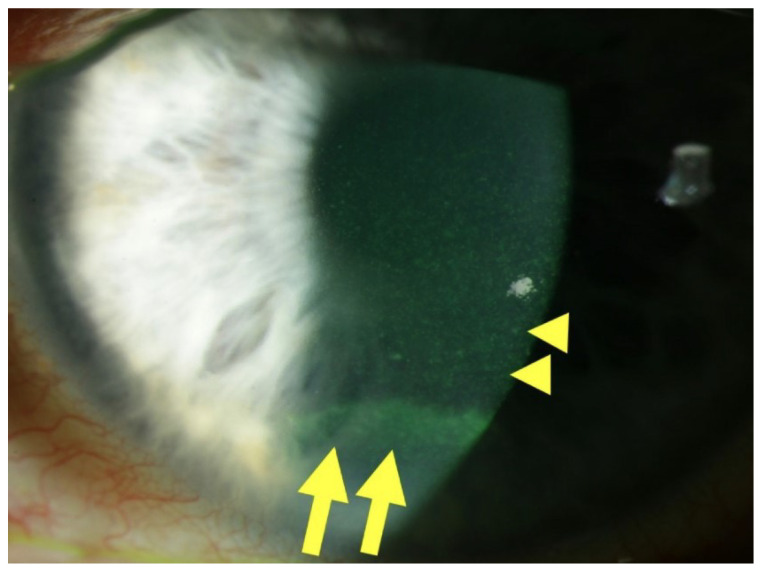
Slit lamp photograph of a patient with diabetic keratopathy. Centrally, fluorescein staining reveals SPKs (yellow arrowheads), as well as an epithelial ridge or pseudo-dendrite, indicative of a recently closed erosion (yellow arrows), in the inferior third of the cornea. From “Diabetic Keratopathy: Redox Signaling Pathways and Therapeutic Prospects” by Buonfiglio, F.; Wasielica-Poslednik, J.; Pfeiffer, N.; Gericke, A. *Antioxidants* **2024**, *13*, 120, Introduction, Figure 1, (https://doi.org/10.3390/antiox13010120 (accessed on the 11 March 2024)). CC BY. [4].

**Figure 2 ijms-25-03471-f002:**
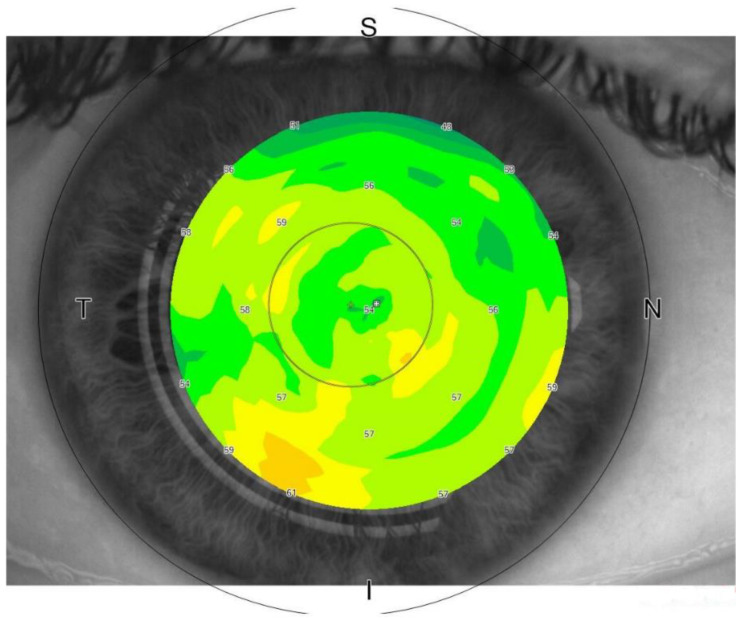
Epithelial thickness maps as shown by a commercially available AS-OCT machine. The thickness of the corneal epithelium can be accurately measured by OCT imaging at the centre and in the peripheral sectors (N: nasal; T: temporal; S: superior; I: inferior). From “Fourier-Domain OCT Imaging of the Ocular Surface and Tear Film Dynamics: A Review of the State of the Art and an Integrative Model of the Tear Behavior during the Inter-Blink Period and Visual Fixation.“ by Napoli, P.E.; Nioi, M.; Mangoni, L.; Gentile, P.; Braghiroli, M.; d’Aloja, E.; Fossarello, M. *J. Clin. Med.* **2020**, *9*, 668. Section 6, Figure 6 (https://doi.org/10.3390/jcm9030668 (accessed on 5 Match 2024)). CC BY. [6].

**Figure 3 ijms-25-03471-f003:**
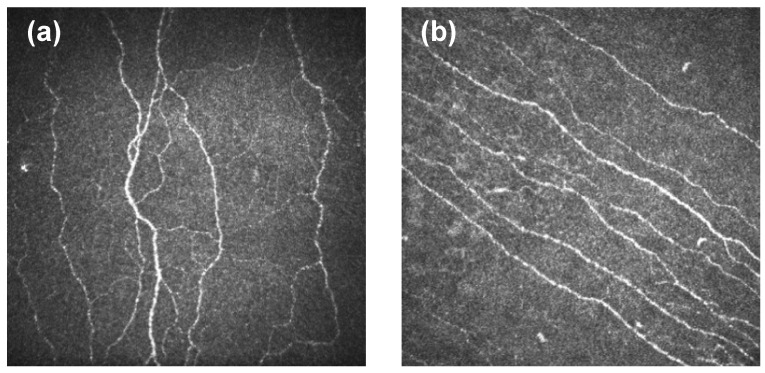
SBNP images of 2 patients acquired via laser-scanning IVCM. (**a**) High tortuosity. (**b**) Low tortuosity. From “New Method for the Automated Assessment of Corneal Nerve Tortuosity Using Confocal Microscopy Imaging.” by Fernández, I.; Vázquez, A.; Calonge, M.; Maldonado, M.J.; de la Mata, A.; López-Miguel, A. *Appl. Sci.* **2022**, *12*, 10450, Introduction, Figure 1, (https://doi.org/10.3390/app122010450 (accessed on the 5 March 2024)). CC BY. [42].

**Table 1 ijms-25-03471-t001:** AS-OCT CCET measurements in diabetic patients vs. controls.

Study	Country	Sample Size (DM vs. Controls)	Device	CCET DM	CCET Controls	Results
D’Andrea L et al., 2022 [5]	Italy	60/60	FD-OCT RTVue100 (Optovue, Inc., Fremont, CA, USA)	55.48 ± 3.67	51.80 ± 4.67	DM thicker in a controlled diabetes population
Yusufoğlu E et al., 2023 [7]	Turkey	72/72	Canon SD HS100 (Canon, Tokyo, Japan)	52.25 ± 3.61	53.95 ± 3.05	DM thinner
Elmekawey H et al., 2020 [8]	Egypt	25/25	FD-OCT RTVue100 (Optovue, Inc., Fremont, CA, USA)	53.36 ± N/A	52.56 ± N/A	Nonsignificant in a cataract population
Gunay M et al., 2016 [9]	Turkey	26/20	FD-OCT RTVue100 (Optovue, Inc., Fremont, CA, USA)	53.7 ± 6.7	54.1 ± 5.9	Nonsignificant in a pediatric population

**Table 2 ijms-25-03471-t002:** CCET measurements from Rosenberg et al., 2000 [10].

	Controls	None	Mild	Severe
Neuropathy	50.7 ± 3.5	51.9 ± 4.6, *p* = 0.55 *	53.2 ± 7.4, *p* = 0.41 *, *p* = 0.70 †	45.3 ± 3.2, *p* = 0.016 *, *p* = 0.017 †, *p* = 0.061 ‡
Nephropathy	50.7 ± 3.5	51.9 ± 5.9, *p* = 0.61 *	N/A	49.5 ± 5.9, *p* = 0.59 *, *p* = 0.39 †
Retinopathy	50.7 ± 3.5	53.0 ± 4.0, *p* = 0.25 *	N/A	49.1 ± 6.4, *p* = 0.50 *, *p* = 0.17 †

* compared to controls, † compared to None subgroup, ‡ compared to Mild subgroup.

## Data Availability

The first author and the corresponding author have full access to all the data and materials in the study.

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
