# Peer review of "Corneal Epithelial Changes in Diabetic Patients: A Review"

_ijms, 2024, doi:10.3390/ijms25063471_

Round 1
Reviewer 1 Report
Comments and Suggestions for Authors
This review will be highly beneficial to the readers. I congratulate the authors for coming up with such a nice piece of work. I advise the authors to include the following suggestions in their manuscript to improve its overall quality.
1. The authors are advised to provide images of diabetic epithelial keratopathy (DEK) at different stages to orient readers.
2. A dedicated section on the specific clinical signs associated with DEK and their relevance to corneal health should be included.
3. The authors should discuss in detail the reasons for corneal edema.
4. The authors have mentioned the XYZ hypothesis. They are advised to concisely discuss its implications for understanding diabetic corneal epithelial changes.
5. The authors are advised to illustrate key concepts of the corneal SBNP and its role in wound healing through proper flowcharts and figures.
6. Please provide more insights on how the neurotrophic states can impact the reliability of corneal epithelial measurements.
Comments on the Quality of English LanguageMinor editing of English language required
Author Response
Dear reviewer,
we have made changes according to your recommendations, which we found very useful. As per point 3, the reasons have been made more clear in the corneal edema section.
Point 6 has received a more detailed phrase alongside a new reference.
We are struggling with point 5 however, as none of the authors is or has access to an illustrator.
Best wishes
Reviewer 2 Report
Comments and Suggestions for Authors
Reviewer’s report
Manuscript title: Corneal epithelial changes in diabetic patients: a review.
Manuscript ID: ijms-2907141
General comments:
The manuscript by Lidia Ladea et al. covers the cornea epithelial changes under the influence of diabetes, mostly on the structural changes, mechanisms, and factors involved in diabetic epithelial keratopathy (DEK), and diagnostic methodologies. Authors also tried to explore the potential use of these structural changes in clinical diagnostics.
The manuscript is well-written and provides comprehensive and updated information. Nevertheless, there are some minor points which demand corrections or revisions, as specified in the minor comments below.
Minor comments:
1. Abstract, line 4- 5: The paper systematically examines alterations in corneal epithelial structure and function observed in diabetic patients.
The manuscript seems not cover too much on the functional aspects of corneal epithelium in diabetic patients. The functional aspects referred in the manuscript are wound healing, cornea sensitivity, serving as a physical barrier, and conferring the biomechanical properties. Cornea sensitivity is more related to cornea nerves, not the epithelium. The biomechanical properties are the traits conferred by the entire cornea, not just the cornea epithelium. It is therefore recommended to revise the sentence as ‘The paper systematically examines alterations in corneal epithelial structure and the impacts related to the alterations observed in diabetic patients.’.
2. Abstract, line 8: ……. looking at factors such as hyperglycemia, oxidative stress, and inflammatory pathways.
There is no direct mention about inflammatory pathways in the manuscript. Authors would need to add information on inflammatory pathways or delete it in the abstract.
3. Abstract, line 10: ‘…….addressing potential links to infections and delayed wound healing….’
There is no direct mention about links to infections in the manuscript. It is suggested to revised as ‘‘…….addressing potential links to heightened proteolytic enzyme activities and delayed wound healing….’.
4. Lines 123, 185, 347: Xu et al. Zou et al. Pérez-Rico et al.
5. Line 151: ……of patients suffering from diabetes …..
6. Line 216: In vivo confocal microscopy is a non-invasive diagnosis tool,…
7. In Figure 2 legend: ‘SNBP’ should be corrected as ‘SBNP’. Also, the labelling in figure is (a), (b), not consistent with the (A), (B) in the figure legend.
8. Line 322: ‘…two measurement devices that can assess corneal biomechanical parameter..’
9. Line 326: ‘….has great repeatability and reproducibility ….’
10. Line 343: ‘…..did not find differences in CH and CRF..’
11. Line 344: ‘……, nor a correlation was found between these parameters….’
12. Line 391: ‘….in this function will manifest a swelling…’
13. Line 405: ‘….., which would show a more generalized thickening.’
14. Line 409: ‘…..would be a variable that would quantify epithelial regularity, rather than simply using CCET.’
Comments on the Quality of English LanguageSome minor revisions and corrections should be endeavored.
Author Response
Dear reviewer,
thank you for the attention with which you have read our work. We have applied all of your suggestions, bar for nr 12 which you could not find any issue with (we indeed do mean "manifest as swelling"). The changes are highlighted in the new version of the manuscript, except for the ones in the Abstract, which i cannot get the latex editor to properly highlight. Minor comments 1, 2 and 3 are errors that have slipped through the editing process after culling of some paragraphs.